# The Level of Burden among Caregivers of Patients with Alzheimer’s Disease in Saudi Arabia

**DOI:** 10.3390/ijerph20042864

**Published:** 2023-02-06

**Authors:** Amal Mohammad. Badawoud, Yasmin K. AlQadheeb, Shahad S. AlZahrani, Razan A. AlGhamdi, Elaf A. Alanazi, Sarah M. AlFozan, Norah S. AlJafer, Ibrahim M. Asiri, Fawaz M. Alotaibi

**Affiliations:** 1Department of Pharmacy Practice, College of Pharmacy, Princess Nourah Bint Abdulrahman University, Riyadh 11671, Saudi Arabia; 2College of Pharmacy, Princess Nourah Bint Abdulrahman University, Riyadh 11671, Saudi Arabia; 3College of Clinical Pharmacy, Imam Abdulrahman bin Faisal University, Dammam 31441, Saudi Arabia; 4Pharmacy Practice Department, College of Clinical Pharmacy, Imam Abdulrahman Bin Faisal University, P.O. Box 1982, Dammam 31441, Saudi Arabia

**Keywords:** Alzheimer’s disease, caregiver burden, medication knowledge, coping techniques

## Abstract

Background: Caregiver burden is a serious global issue associated with the growing number of older adult patients with Alzheimer’s disease (AD). AD patients become more dependent on their caregivers and require assistance with basic daily life activities. This study aims to measure the caregiver burden of informal caregivers of AD patients and to determine their characteristics. In addition, it intends to understand caregiver coping techniques and assess their medication knowledge. Methods: This was a cross-sectional study including 148 informal caregivers mainly recruited by the Saudi Alzheimer’s Disease Association (SADA). A four-part study questionnaire was used for data collection in the Arabic Language and included the following: socio-demographic characteristics of AD patients and their caregivers, the 12-item version of the Zarit Burden Interview (ZBI), and adapted questions on coping techniques and medication knowledge. Results: A total of 148 caregivers (62% were female) participated in this study, and 79.06% were between 30 and 60 years old. The ZBI average score was 27, indicating a moderate to high burden. Caregivers reported their need for services to improve their quality of life. The medication knowledge was insufficient in most aspects except that more than half were aware of medications’ side effects. Conclusion: Our study revealed that the average burden among informal caregivers of AD patients was moderate–high.

## 1. Introduction

Population aging is the leading demographic phenomenon in the 21st century [1]. In 2020, there were 727 million individuals aged 65 years and older worldwide, and this number is expected to double over the next three decades [2]. In Saudi Arabia (SA), according to a preliminary estimates report in mid-2020, 3.2% of the population was 65 years and older. The older people in SA are following the growing aging trends worldwide; they are expected to reach 9.3% of the population by 2030 and 28.2% by 2050 [3]. Therefore, as the older population increases, an increase in aging-associated diseases, such as Alzheimer’s, is expected.

AD is not a part of normal aging. However, after the age of 65, the incidence of developing AD doubles every five years on average [4]. Globally, over 50 million people were living with dementia in 2020 [5]. It has been estimated that the number of AD patients will double in the next 20 years, reaching 82 million in 2030 and 152 million in 2050 [5]. Likewise, there are approximately 130 thousand AD patients in SA, which is expected to increase to nearly triple to 14 million by the year 2060 [4]. It is a neurodegenerative disease characterized by a progressive decline in cognitive function. AD substantially affects older people, causing a progressive decline in memory, thinking, language, and learning capacity [6]. AD has a negative impact on the patient’s life, career, family, and social patterns. Moreover, it is a serious condition that could worsen over time if left untreated and may potentially lead to death [7].

In current practice, there is no cure for AD. However, cognitive, behavioral, and psychological symptoms of AD can be optimally controlled with both nonpharmacologic and pharmacologic interventions. Therefore, as the disease advances, AD patients become more dependent on their caregivers and require assistance with basic daily life activities.

Previous studies have reported that family members are the primary source of caregivers for Alzheimer’s patients [8]. As a result, those caregivers are facing daily stressors most of the time that are either ignored or dealt with it through different coping techniques. However, the concept of coping mechanisms varies among individuals due to government resources, economic status, and cultural differences [9]. In addition, medication management is one of the most important tasks for informal caregivers in community care settings. Therefore, for caregivers to have a vital part in managing medications for their relatives, medication knowledge must be improved [10].

Furthermore, the lack of provision of dementia services within the public healthcare system negatively impacts both AD patients and their caregivers [11]. Studies concerning caregiver burden and the necessities requested by care providers are consistent with the Saudi health transformation strategy (2030). The strategy confirmed inadequate capacities in extended care services and elaborated on overcoming that, and this will positively influence older people requiring home care and long-term services. The strategy also comprehensively detects population needs to successfully allocate resources [12]. In addition, the “keep well” model focuses on educating and empowering patients and their caregivers through multiple coaching programs, which addresses the current issue of care providers requesting educational programs. Moreover, the strategy also works on attaining a new model of care revolving around compassionate approaches, which is crucial to the well-being of dementia patients and informal care providers [12].

A study conducted in the U.S. assessed the level of burden among informal caregivers of AD patients. It showed a considerable variation in the burden levels ranging from no to too severe a burden [13]. Whereas in SA, three studies conducted in different cities showed that most family caregivers felt a moderate burden [14,15,16].

Family caregivers often spend effort and time to meet all the health and personal needs of AD patients. Still, they often neglect their own physical and psychological health in the process. Therefore, caregivers report high levels of stress, which may consequently lead to various diseases and issues. Coping techniques help caregivers in adapting and meet the demands of caregiving [17]. A study stated that the majority of caregivers of mental disorder patients in SA relied on religious acts and spirituality as effective coping mechanisms. Some also described it as a mercy from God and found acceptance to be a helpful approach to cope with the burden [18]. However, some Saudi caregivers are unaware of the importance of using effective techniques due to their limited knowledge and awareness and lack of mental health literacy [19].

A qualitative study was conducted in Jeddah reflecting the experiences of 21 primary caregivers of patients using antipsychotic medication. The responses revealed a negative attitude toward medication use. There was also limited access to obtain valid information about medications and strategies to avoid side effects. Moreover, improving medication knowledge might allow some caregivers to feel active in providing the best medication management for their relatives [10]. Therefore, more information resources are required for this role, which requires specific medication management skills and knowledge.

The demands of caregiving can limit not only caregivers’ ability to take care of their patients but also how they take care of themselves. Hence, identifying local support organizations such as the local Alzheimer’s Association helps caregivers overcome challenges [20]. Furthermore, caregivers’ ability to manage daily caregiving difficulties and stress can be enhanced with education and skill training [21]. The first local Alzheimer’s Association in SA is the Saudi Alzheimer’s Disease Association (SADA). SADA provides many services and support for caregivers, such as answering their questions through the helpline and reading materials. It also conducts meetings, training sessions, and educational events for caregivers of AD patients and healthcare professionals. However, the medical healthcare services that should be provided for AD patients and their caregivers are still limited. For instance, there is a demand for services such as psychotherapy, patient self-help groups, and psycho-education groups [22].

Therefore, the primary aim of this study was to determine the caregiver burden of AD patients in a sample of the Saudi population and the characteristics of their informal caregivers. The secondary outcomes were to explore the coping techniques of the caregivers of AD patients and to determine the medication knowledge of caregivers of AD patients by each gender.

## 2. Materials and Methods

### 2.1. Study Design

This was a cross-sectional study that targeted adult, informal caregivers of older adults with Alzheimer’s disease in the Kingdom of Saudi Arabia. We recruited informal caregivers (without regard to nationality) who are providing unpaid care and assistance physically or emotionally to an older adult individual. In our study, we included any informal caregiver 18 years or older, currently caring for an AD patient at any stage (mild, moderate, or severe), and able to communicate in Arabic to complete the questionnaire, which was primarily in Arabic.

### 2.2. Data Collection Tool

This study was based on a survey developed via QuestionPro, a publicly free platform to create a questionnaire for different purposes. The survey included 4 sections, which include the following: socio-demographic information (age, sex, educational level, marital status, relationship with the AD patients, duration of care, and other questions related to caregivers) and the short version of Arabic version of the Zarit Burden Interview (ZBI) tool. In addition, coping mechanisms are very important to be applied in order to control different stressors. As a result, we have created set of questions to identify the source of coping techniques each of the caregivers needs and performs. All the questions have been implemented from previous studies that were published on coping mechanisms [17,18].

The coping mechanism was determined by asking the participants about the strategies used to overcome difficulties they encounter while providing care, availability of resources needed to provide care, services to improve quality of life, and support to reduce the burden of care. Additionally, participants were asked about the source they used to obtain information about AD.

In addition, questions regarding medication knowledge were included to test the participants’ knowledge regarding each medication taken. All those sections are validated surveys and have been adopted from previous studies. 

The short version of ZBI tool was used to assess the burden among the participants. It is one of the most employed caregiver burden measures used in the literature. The ZBI tool originally consists of 22 items. However, a comprehensive shorter version has been developed and validated to provide results equivalent to the full version. Therefore, reducing the number of items did not affect the quality of the ZBI tool while making the instrument more convenient to administer [23]. The short version includes 12 items, which are briefly listed in Figure 1. The 12-item, self-reported version of ZBI assesses burden associated with behaviors and relationships of caregivers with individuals who they are taking care of. The responses are scored using a five-point Likert scale: never (0), rarely (1), sometimes (2), quite frequently (3), and nearly always (4).

Moreover, the psychometric characteristics including validity and reliability of the developed survey were tested on through several steps. First, we conducted face validity by sending the survey to five experts in the field of geriatrics research. Second, we conducted our pilot study to ensure the survey accuracy of the targeted participants. We made sure that those who were included in the pilot study were removed from the study analysis. Finally, Cronbach Alpha was performed to test the reliability (0.78) of the final version of the Arabic ZBI tool [15].

Our targeted sample size was calculated using Raosoft online calculator (Raosoft Inc., Seattle, WA, USA) based on the Saudi AD patients during the conducting of the study. We found that we needed 385 participants to gain a power with 95% confidence interval. After the survey development process was completed, we started distributing the survey through a collaboration with Saudi Alzheimer’s Disease Association, an association that cares about older adult population suffering from AD. This association helped us in terms of obtaining access to the patients by distributing the survey to caregivers via email and other social media platforms after signing the consent form to complete the study. The Institutional Review Board (IRB) was granted from both institutions, Princess Nourah Bint Abdurrahman University (PNU) and Imam Abdulrahman Bin Faisal University (IAU).

### 2.3. Statistical Analysis

The questionnaire was analyzed using SPSS.IBM, version 19 [24]. Mean and standard deviation were reported for the continuous variables, while the categorical variables were reported by calculating the frequency and percentage. After collecting all the variables, assumptions were checked to provide us with the proper prediction module that should be used. We then selected a cutoff for the ZBI total score to classify it as a high or low burden. We defined a high burden as a ZBI score ≥ 24, while a low burden was a ZBI score < 24. The crude and adjusted logistic regression methods were used to investigate the relationship between caregiver burden and coping strategies. Moreover, all the variables were tested at a 0.05 level of significance.

## 3. Results

A total of 345 participants accessed the survey. However, it was completed by 148 participants with a completion rate of 42.89%. The study recruited 92 (62%) female caregivers and 56 (38%) male caregivers. Most of the participants (79.06%) were between the age of 30 to 60 years, and more than half (58.78%) currently reside in the central province. Additionally, a considerable number of caregivers (58.11%) were educated with a bachelor’s degree, with 54.05% being employed and 63.51% being married. The caregivers taking care of Alzheimer’s patients for the first time comprised 85.14% of the sample, with 71.62% being first-degree relatives of the patients. Regarding the time spent taking care of the patients, 62.16% spend more than 4 h daily tending to the needs of Alzheimer’s care recipients. Moreover, 43.24% of the caregivers were taking care of Alzheimer’s patients for 4 to 9 years followed by 40.54 (%) of caregivers taking care of the patients for three years or less. The average ZBI score among the participants was 27 (±SD), which signifies a moderate–high level of burden among caregivers. (Table 1).

The AD patients included in the study were 83 (56.08%) males and 65 (43.92%) females with the majority (96.62%) being older than 60 years. Almost half of the patients (49.32%) followed up in a government health facility. Additionally, 32.43 (32.43%) of the patients were in the last stage of AD, and more than half of the study sample (66.89%) had other chronic diseases. (Table 2). Figure 1 shows the answers to the caregiver burden assessment that represent the feelings of a caregiver in providing care for Alzheimer’s patients.

The study contained seven domains to assess coping techniques among caregivers of AD patients, including ways to overcome difficulties they encounter while providing care, the adequacy of resources (including financial, healthcare, and mental health), and the sources of information and support. The results of all the domains—considering males and females—were not significant except for the need for services to improve the quality of the caregivers’ and patients’ lives where it was significant (*p*-value 0.02). The difference in the type of services preferred by males and females is that more males prefer to have a center dedicated to AD patients; however, the difference analysis shows that more females prefer to have provided courses or programs on AD and how to use them than males. (Table 3).

The participants’ knowledge regarding the care recipients’ medications was poor in most aspects; 72% of caregivers were unable to list all the medications taken by their patients (*p*-value 0.46), and 76% did not have enough knowledge regarding all the indications of their patient’s medications (*p*-value 0.64). Moreover, most caregivers were unaware of how the medications are supposed to be taken and when to administer each medication (84% and 86%, respectively) (*p*-value 0.01 and 0.39, respectively). Meanwhile, more than half of the participants (62%) were aware of the medications’ side effects (*p*-value 0.58), and 66% knew what needed to be conducted if a side effect ever occurred (*p*-value 0.73). In case of a missed dose, only 19% out of all the caregivers knew what action needed to be taken (*p*-value 0.25). (Table 4).

We further investigated the association of burden with the coping strategies adopted by caregivers. In the crude analysis, participants who answered yes to owning all the resources or having some resources, respectively, were significantly less likely to experience a high burden than those who answered no to having any resources (OR: 0.16 (95% CI, 0.04–0.53), *p*-value < 0.01; OR: 0.32 (95% CI, 0.11–0.94), *p*-value 0.03, respectively). Participants were found to be less likely to experience a high burden when asking specialized doctors than those who were seeking social media and other platforms for information about taking care of AD patients and themselves (OR: 0.30 (95% CI, 0.14–0.65), *p*-value < 0.01). Further, caregivers who feel that they receive adequate support for AD care had less burden compared to those not receiving adequate support (OR: 0.23 (95% CI, 0.11–0.47), *p*-value < 0.01). In addition to the crude analysis for whether or not adequate support was received, the results of the adjusted analysis were consistent (OR: 0.29 (95% CI, 0.11–0.73), *p*-value < 0.01). In contrast, participants who received community support were more likely to experience a high burden (OR: 5.28 (95% CI, 1.15–24.04), *p*-value 0.03) (Table 5).

## 4. Discussion

This is a study conducted to investigate the level of burden and its association with coping techniques and medication knowledge among caregivers of AD patients in SA. To the best of our knowledge, this is the first study that touched on this topic in SA.

Caring for a person with AD can be stressful and challenging, especially for informal caregivers. The findings of this study determined the level of burden among caregivers of AD patients in SA. The results indicated that caregivers felt a moderate to high burden with a mean of a 27.87 level of burden. This is consistent with previous studies that have been conducted in different cities in SA [14,15,16]. The 12-item version of the ZBI tool was used, and the caregivers’ feelings (such as stress, anger, not having enough time, and quality of life) were the key to answering all the domains. The significant factor associated with caregiver burden in the Najran study [15] was age, stating that the burden increased as age increased. Nevertheless, most of the caregivers in our study fell between the 40–50s age category. Therefore, we could not assess such an association.

The majority of our study participants were female caregivers, those with a bachelor’s degree, employees. This is consistent with previous studies that were conducted in Jeddah and Riyadh except for the employment statuses; for the study conducted in Riyadh, the majority were unemployed [14,16]. Moreover, in our study, most of the caregivers were first-degree relatives, which is consistent with the previous studies conducted in Riyadh and Najran [14,15].

The main characteristics of AD patients in this study were male, over 80 years old, having multiple comorbidities, and this is consistent with the previous study completed in Riyadh [14]. In addition, our study assessed the type of health establishment the AD patients in SA refer to and found that more than half of the patients follow a governmental health facility.

In Table 5, we have found an association between certain coping techniques/resources and the burden of caring for AD patients. Participants who obtained support from the community were more likely to feel the burden than those who received support from family members. These interesting results could be due to the nature of Saudi families who favor/expect support from the household more than any other type of support. Thus, a support program is very much needed for family caregivers who care for multiple generations (the oldest old and children) in the same house. In addition, participants who seek to obtain information from a specialized doctor feel less likely to have a higher burden, which emphasizes the point that always acquiring medical information from trusted resources improves the quality of the patient’s life. To the limited local data, we could not perform any comparison to our results.

In the same manner as the study conducted in Jeddah [10], caregiver medication knowledge was still lacking, which ultimately reduces patients’ welfare. Therefore, educating caregivers adequately concerning medications can improve AD patients’ quality of care and reduce caregiver burden. In addition, conducting medication review sessions and creating a medication list including all medication names and administration directions as well as information about the patient, caregiver healthcare providers, and pharmacy would enhance safe medication use by caregivers. The results of this study showed that most caregivers relied heavily on family support, which could be related to the Islamic and cultural aspects of the Saudi community. This is consistent with data from a previous study that showcased the significance of spirituality as a coping technique in religious communities [18]. This finding highlighted the importance of examining societal characteristics before planning caregiver-centered programs, which are important to address caregiver needs. In addition, faith-based approaches should be particularly considered in SA, such as Islamic beliefs and practices.

Although some previous studies assessed caregiver burden in caregivers of AD patients, those studies had a generalizability issue because each study was conducted in a single city in SA [14,15,16]. The aim of our study was to assess the caregiver burden of AD patients all around SA. Moreover, we successfully included participants from the three main provinces in SA (central, eastern, and western) and this was achieved with the collaboration of the Saudi Alzheimer’s Disease Association. Therefore, the results of this study can be used to describe the caregivers of AD patients in these Saudi provinces. In addition, this study looked at several important caregiving aspects for patients with AD, including determining burden level, medication knowledge, and coping techniques. Our study was the first study using the Arabic short version of the ZBI (12 items). Moreover, the questions regarding medication knowledge and coping techniques were adapted from published studies [10,17,18].

This study has some limitations. First, we could not include enough participants compared to the calculated sample size, as the number of respondents was 148 out of 345 recipients, which affected the study’s generalizability. This was because the period of conducting the study was not long enough to collect more responses. The second limitation was the sensitivity of the questions of the ZBI tool. According to the recipients’ number versus the responders’ number, we could state that the sensitivity of questions contributed to the low response rate. Third, the survey was too long. Although the short version of the ZBI was used, the length of the survey could not be avoided since it included four main domains: the level of burden, medication knowledge, and coping techniques, as well as the patients’ and caregivers’ characteristics. In addition, the cutoff point for the short version of the ZBI tool was determined arbitrarily. However, we tried our best to select the most validated version with high sensitivity and specificity [23].

## 5. Conclusions

Our study revealed that the average burden among the informal caregivers of AD patients was moderate–high. This outlines the need to explore ways to alleviate caregiver burden and further enhance patient care. On the other hand, our findings emphasized the importance of implementing programs and sessions to educate and train caregivers on managing the difficulties arising from their role.

## Figures and Tables

**Figure 1 ijerph-20-02864-f001:**
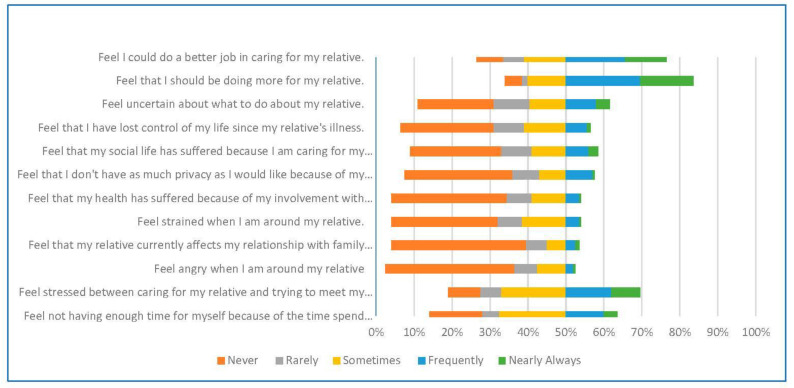
Caregiver Burden Assessment that Represents the Feelings of a Caregiver in Providing Care for Alzheimer’s Patients.

**Table 1 ijerph-20-02864-t001:** Baseline Demographics and Characteristics of Caregivers for Alzheimer’s Patients.

Characteristics	n (%)
**Age**	
18–29 years	25 (16.89)
30–40 years	34 (22.97)
41–50 years	45 (30.41)
51–60 years	38 (25.68)
61 or older	6 (4.00)
**Gender**	
Female	92 (62.16)
Male	56 (37.84)
**Educational level**	
Less than high school	9 (6.08)
High school	28 (18.92)
Bachelor’s degree	86 (58.11)
Postgraduate degree	25 (16.89)
**Employment Status**	
Employed	80 (54.05)
Not Employed	68 (54.95)
**Marital status**	
Single	44 (29.73)
Married	94 (63.51)
Divorced/widowed	10 (6.76)
**Relationship with the Alzheimer’s patient**	
1st-degree relative	106 (71.62)
2nd-degree relative	23 (15.54)
Informal non-relative	19 (12.84)
**First time taking care of an Alzheimer’s patient**	
Yes	126 (85.14)
No	22 (14.86)
**Number of hours spent per day taking care of Alzheimer’s patient**	
Less than 2 h	16 (10.81)
2–4 h	40 (27.03)
More than 4 h	92 (62.16)
**Length Taking Care of the Alzheimer’s Patient**	
≤3 years	60 (40.54)
4–9 years	64 (43.24)
≥10 years	24 (16.22)
**Current province living in in Saudi Arabia**	
Central province	87 (58.78)
Eastern province, Northern province, and Others	14 (9.46)
Western province and Southern province	47 (31.76)
**Zarit Burden Interview (ZBI) for Caregiver Burden Assessment**	**Mean (±SD)**
	27.87 (8.79)

**Table 2 ijerph-20-02864-t002:** Baseline Demographics and Characteristics of Alzheimer’s Patients Answered by the Caregivers.

Characteristics	n (%)
**Age**	
51–60	5 (3.38)
61–70	30 (20.27)
71–80	48 (32.43)
>80	65 (43.92)
**Gender**	
Male	83 (56.08)
Female	65 (43.92)
**Type of Health Establishment**	
Government health facility	73 (49.32)
Private health facility	29 (19.59)
Both	31 (20.95)
Other	15 (10.14)
**Stage of Alzheimer’s Disease**	
I don’t know	49 (33.11)
Yes, first stage	20 (13.51)
Yes, second stage	31 (20.95)
Yes, last stage	48 (32.43)
**Time Since Alzheimer’s Disease Diagnosis**	
<5 years	69 (46.62)
5–10 years	61 (41.22)
>10 years	18 (12.16)
**Other Chronic Diseases**	
I don’t know	10 (6.76)
Yes, he has other chronic diseases	99 (66.89)
No, he doesn’t have any other chronic diseases	39 (26.35)

**Table 3 ijerph-20-02864-t003:** Information on Coping Strategies and Support for Caregivers of Alzheimer’s Patients.

Question	Female n (%)	Male n (%)	*p-*Value
**What do you do to overcome the difficulties you encounter while providing care?**			
I do nothing	16 (59.26)	11 (40.74)	0.36
I talk to people who have a similar interest and/or have had previous experience caring for Alzheimer patients	36 (50.00)	36 (50.00)	
Attend courses or take behavioral therapy sessions to learn about Alzheimer’s disease and how to deal with it	15 (71.43)	6 (28.57)	
Other	16 (57.14)	12 (42.86)	
**Do you think you have all the resources (including financial, health care, and mental health) that you need as an Alzheimer’s caregiver?**			
Yes, I own all the resources	19 (63.33)	11 (36.67)	0.86
Yes, I have some resources	52 (60.47)	34 (39.53)	
No, I don’t have it	21 (65.63)	11 (34.38)	
**What services do you need to improve the quality of your life and that of your relative “Alzheimer’s patient”?**			
Providing courses or programs on Alzheimer’s disease and how to deal with it	24 (77.42)	7 (22.58)	0.02 *
Provide behavioral therapy sessions to caregivers	9 (56.25)	7 (43.75)	
Center dedicated to Alzheimer’s patients	32 (44.44)	40 (55.56)	
Providing (medical and health) supplies for Alzheimer’s patients	18 (37.93)	11 (37.93)	
**From what source do you receive information about Alzheimer’s disease and how to care for yourself and an Alzheimer’s patient?**			
Specialized Doctors	37 (55.22)	30 (44.78)	0.96
Medical resources	10 (58.82	7 (41.18)	
Social Media other platforms	36 (56.25)	28 (43.75)	
**Often, from whom do you receive support (financial or moral) during your Alzheimer’s care to reduce the burden on you?**			
Government support	13 (59.09)	9 (40.91)	0.94
Community Support	12 (63.16)	7 (36.84)	
Family Support	67 (62.62)	40 (37.38)	
**Do you have a desire to enroll in courses to learn more about how to overcome the psychological, financial, and physical burden while caring for an Alzheimer’s patient?**			
Yes	69 (62.73)	41 (37.27)	0.8
No	23 (60.53)	15 (39.47)	
**In general, do you feel that you are receiving adequate support for Alzheimer’s care?**			
Yes	34 (64.15)	19 (35.85)	0.7
No	58 (61.05)	37 (38.95)	

* *p*-Value < 0.05.

**Table 4 ijerph-20-02864-t004:** Caregiver Knowledge Regarding Medications for Alzheimer’s Patients.

Question	Entire Participants n (%)	Female n (%)	Male n (%)	*p*-Value
**Can you list the names of all the medications used by the Alzheimer patient you are taking care of?**				
No	107 (72)	62 (57.94)	45 (42.06)	0.46
Yes	41 (28)	21 (51.22)	20 (48.78)	
**Do you have sufficient knowledge of the indications for each medication used by the Alzheimer’s patient you are taking care of?**				
No	112 (76)	64 (57.14)	48 (42.86)	0.64
Yes	36 (24)	19 (52.78)	17 (47.22)	
**Do you know how the Alzheimer’s patient is supposed to take each of his medications? (In terms of specific dosage, how to use: if it is taken by mouth or by subcutaneous needle, and directions after eating or before)**				
No	124 (84)	75 (66.48)	49 (39.52)	0.01 *
Yes	24 (16)	8 (33.33)	16 (66.67)	
**Do you know when to give each medication to the Alzheimer’s patient you care for?**				
No	127 (86)	73 (57.48)	54 (42.52)	0.39
Yes	21 (14)	10 (47.62)	11 (52.38)	
**Do you know what are the side effects of the medications taken by the Alzheimer’s patient you are taking care of?**				
No	92 (62)	50 (54.35)	42 (45.65)	0.58
Yes	56 (38)	33 (58.93)	23 (41.07)	
**Do you know what to do if an Alzheimer’s patient experiences “side effects” from the medications they are taking?**				
No	98 (66)	54 (55.10)	44 (44.90)	0.73
Yes	50 (34)	29 (58.00)	21 (42.00)	
**Are you aware of what you do if you forget the scheduled dose for an Alzheimer’s patient’s medication?**				
No	120 (81)	70 (58.33)	50 (41.67)	0.25
Yes	28 (19)	13 (46.43)	15 (53.57)	

* *p*-Value < 0.05.

**Table 5 ijerph-20-02864-t005:** Association of the Burden with the Coping Strategies Adopted.

Question	Crude OR (95% CI)	*p*-Value	Adjusted OR (95% CI)	*p*-Value
**What do you do to overcome the difficulties you encounter while providing care?**				
I do nothing	0.95 (029–3.04)	0.93	0.96 (0.24–3.69)	0.95
I talk to people who have a similar interest and/or have had previous experience caring for Alzheimer patients	0.52 (0.20–1.35)	0.18	0.59 (0.19–1.76)	0.34
Attend courses or take behavioral therapy sessions to learn about Alzheimer’s disease and how to deal with it	1.28 (0.35–4.68)	0.7	1.19 (0.25–5.64)	0.82
Other	Ref.	Ref.	Ref.	Ref.
**Do you think you have all the resources (including financial, health care, and mental health) that you need as an Alzheimer’s caregiver?**				
Yes, I own all the resources	**0.16 (0.04–0.53)**	**<0.01**	0.53 (0.12–2.36)	0.41
Yes, I have some resources	**0.32 (0.11–0.94)**	**0.03**	0.48 (0.14–1.65)	0.24
No, I don’t have it	Ref.	Ref.	Ref.	Ref.
**What services do you need to improve the quality of your life and that of your relative “Alzheimer’s patient”?**				
Providing courses or programs on Alzheimer’s disease and how to deal with it	0.50 (0.17–1.41)	0.19	0.64 (0.18–2.30	0.49
Provide behavioral therapy sessions to caregivers	0.78 (0.22–2.71)	0.7	0.85 (0.18–3.97)	0.84
Center dedicated to Alzheimer’s patients	1.97 (0.78–4.99)	0.14	2.66 (0.87–8.16)	0.08
Providing (medical and health) supplies for Alzheimer’s patients	Ref.	Ref.	Ref.	Ref.
**From what source do you receive information about Alzheimer’s disease and how to care for yourself and an Alzheimer’s patient?**				
Specialized Doctors	**0.30 (0.14–0.65)**	**<0.01**	**0.40 (0.16–0.98)**	**0.04**
Medical resources	0.51 (0.16–1.63)	0.25	0.48 (0.12–1.81)	0.27
Social Media and other platforms	Ref.	Ref.	Ref.	Ref.
**Often, from whom do you receive support (financial or moral) during your Alzheimer’s care to reduce the burden on you?**				
Government support	0.89 (0.35–2.28)	0.82	1.66 (0.54–5.02)	0.36
Community Support	**5.28 (1.15–24.04)**	**0.03**	2.80 (0.54–14.31)	0.21
Family Support	Ref.	Ref.	Ref.	Ref.
**Do you have a desire to enroll in courses to learn more about how to overcome the psychological, financial, and physical burden while caring for an Alzheimer’s patient?**				
Yes	1.49 (0.70–3.18)	0.29	1.17 (0.45–3.05)	0.73
No	Ref.	Ref.	Ref.	Ref.
**In general, do you feel that you are receiving adequate support for Alzheimer’s care?**				
Yes	**0.23 (0.11–0.47)**	**<0.01**	**0.29 (0.11–0.73)**	**<0.01**
No	Ref.	Ref.	Ref.	Ref.

Abbreviations: OR, odds ratio; CI, confidence interval. Statistically significant results are bolded.

## Data Availability

Data are stored with no identifications with the research supervisors.

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
