# Peer review of "The Level of Burden among Caregivers of Patients with Alzheimer’s Disease in Saudi Arabia"

_ijerph, 2023, doi:10.3390/ijerph20042864_

Round 1
Reviewer 1 Report
This paper addresses issues related to the caregivers of people with dementia. The introductions provides a sufficient rationale for the investigation , but the methods and the results are not presented adequately. Unfortunately I had to decide to reject the paper as it stands. There are several points to be emended in the analysis and in the presentation.
I point out only the most important points:
· Introduction page 2 lines 54-56. The two sentences from “Caregivers face…” to “… due to cultural differences” are hard to understand.
Materials and Methods.
· Exclusion criteria are the opposite of inclusion criteria. You do not need to specify that. In this case inclusion criteria are enough.
· The general rationale of the coping strategies investigated is not provided. The authors refer to research in the field of mental disorders, but there are differences between caregiving of the two types of people (for instance, age).
· What the 12 items of the ZBI short version investigate? They should be explained in the Methods.
· The authors speak of the approval of the review board, but they do not mention informed consent from the subjects. This may not be necessary in Saudi Arabia. This should be stated.
Results
· There are many mistakes in the tables, with wrong percentages in Table 1 and particularly in Tables 3 and 4.
· If the focus is the level of burden, this is insufficiently investigated. The authors just give the global mean, whereas they could analyse differences according to other characteristics. Moreover, they do not investigate any association of the burden with the coping strategies adopted.
Discussion
· In the Discussion, the authors state that they did not find any correlation between caregiver’s age and burden. Where is this finding shown?
· Some sentences need to be better explained: see lines 200-201 page 7, or line 204 (the nature of aging?).
Minor points:
Some typos (inconsistence instead of inconsistent, electrically instead of electronically).
Reviewer 2 Report
Abstract
The study describes the level of caregiver burden among caregivers of people with Alzheimer's disease, the characteristics of the dyad, coping strategies and caregivers' knowledge of medication. The study presents a descriptive cross-sectional design developed in the context of Saudi Arabia. Information was collected through a computer-administered survey of a group of people who were made aware of the survey through social networks and/or an association of people with Alzheimer's disease (AD). The study mainly describes demographic variables and the burden score, while the variables relating to coping and knowledge of medication present several response options which are described and analysed by means of a non-described statistical test. The results present the characteristics of the respondents, the level of burden, their coping strategies and the level of knowledge about their relative's medication. The information on coping and medication seems to have been obtained through ad hoc questions from studies conducted in 1995 and 2008. This could affect the validity of the information.
Considerations
It is suggested that the authors frame their research in a stress model, since many of the concepts addressed do not start from a previous definition or a model that supports the relationships between the variables studied.
As a proposal, it is suggested that they review the model of Lazarus and Folkman (Transactional Theory of Stress), in this model a series of contextual variables are contemplated, such as culture, coexistence and many others that may be of interest to them. It also takes into account the stressors to which the caregiver is subjected.
It would be pertinent for them to provide a conceptual definition of the variables they study.
Stratification of the level of burden according to the stage of AD or according to the level of dependency of the person affected by AD is suggested. Another aspect to be taken into account in the assessment of burden is the previous experience of the carer, their sample contains a large proportion of people who are new to caregiving (85.14%) and this may be related to the fact that they experience a higher level of burden.
It is suggested that the authors define more precisely where the sample comes from. Whether the people who participated in the survey were members of the association or what proportion of them were.
It is suggested that the researchers provide reference to the abbreviated version of the ZBI that has been validated in the Arabic-speaking population. The authors refer to an abbreviated scale of the ZBI which is not referenced on line 124. They then refer to the use of a scale which is not specified as having been validated in the Arabic-speaking population (Reference 23; line 132). If possible, please present the information in an organised and unified way. The repetition of information in different paragraphs is unclear and confusing. The Cronbach Alpha value would be very relevant.
It is suggested that the authors take into account the existence of instruments that allow coping strategies to be reliably measured in the caregiving population. There are previous studies where linguistic adaptations have been carried out (https://journals.sagepub.com/doi/full/10.1177/0733464820920100?casa_token=ZIWUnBJdDBgAAAAA%3AyukN2to1wY1qYhH33hX1pExH6VrSPiL37hUHcNCMrFgbtVKzQiCecly3pilt96dnwMfwYPZvlqdkBw). Obtaining information through non-validated instruments could lead to information bias. Was the validity and reliability of the final questionnaire for the measurement of coping and medication knowledge analysed? If so, they could provide information on this.
It is suggested that the authors consider the following aspect. The number of "non-responses" obtained in the questionnaire may limit the representativeness of the sample and its results. It is suggested that the confidence intervals of the measures be calculated in a way that allows the assessment of the precision of the study in the estimation of the measures (as no information is provided on the power of the study to detect association between the groups they compare) and that given the limitations of the representativeness of the sample it is suggested that the authors moderate the claims of the study and limit them to the study participants (characteristics of the sample).
In lines 154 and 155, it is suggested that the researchers provide the percentage figure to which they want to refer (only % appears). The same applies to line 167.
It is suggested that the authors correct the sentence that begins on line 203 and ends on line 204, as the authors do not indicate the performance of a bivariate analysis between the level of caregiver burden and the different socio-demographic variables that would allow the variables to be correlated.
The discussion section lacks an explanation of the results found in the coping strategies investigated.
It is suggested to the authors that in order to indicate that their results are generalisable to the Saudi community because they have collected participants from other provinces, they should first ensure that their sample presents similar characteristics in the variables related to the variables studied (not only in gender, education, degree of kinship and employment). As noted above, the response rate was below the estimated response rate and the characteristics of the resulting sample could be different in relevant variables. This issue appears as a limitation of the study but the statements made throughout the manuscript do not seem to take it into account.
Reviewer 3 Report
1. Line 112, among 148 caregivers, are they migrants from other countries? Add this information. What are their nationalities?
2. 118, electrically is electronically?
3. 120, why sent twice electrically and on social media. What are the differences?
4. 132 Describe the 12 items used in the shortened ZBI
5. 166 The results of ZBI should be presented in more details including the results of each of the 12 items studies and discussed in Discussion.
6. 174 Table 3 is the important findings of this study. The author should describe more details of the results and discuss useful coping strategies in Discussion.
7. 181 Table 3 and 193 Table 4 compares between Male and Female care givers. But this was not mentioned as the objectives of this study. The author needs to be clear in the data/result presentation strategies which support the research purpose.
Round 2
Reviewer 1 Report
Although I acknowledge the authors' effort to improve the paper, I am sorry to say that it has not improved enough and I can not recommend itspublication. Just to mention the most important limitations: 1. the variables included in the model adjusting the estimates of the association between burden and coping strategies are not described in the Methods or in the related table. 2. moreover, most of the associations which were statistically significant when not adjusted were non-significant once adjusted, but the authors mention the creude estimates, and do not comment the effect of the adjustment; 3. the study is now described as a retrospective cohort, whereas this is a cross-sectional study, although it investigates characteristics, habits and strategies that lasted over time.
My suggestion for a new paper is to reduce the number of variables investigated, present findings for males and females with column percentages, clarify the model, consider the adjusted estimates, and comment the differences between crude and adjusted associations.
Reviewer 3 Report
131: clarify the study literatures
134: clarify the study literatures
142: Table 2 does not clearly show which the twelve items are. Please describe clearly.
There are several typos, in particular newly added paras. Examples are below. There seems several others. Please carefully review and correct.
59: economical >> economic
274: kingdome >> kingdom
302: support program >> a support program
